# The Aerobic Denitrification Characteristics of a Halophilic *Marinobacter* sp. Strain and Its Application in a Full-Scale Fly Ash-Washing Wastewater Treatment Plant

**DOI:** 10.3390/microorganisms13061274

**Published:** 2025-05-30

**Authors:** Mengyang Guo, Kai Liu, Hongfei Wang, Yilin Song, Yingying Li, Weijin Zhang, Jian Gao, Mingjun Liao

**Affiliations:** 1Hubei Key Laboratory of Environmental Geotechnology and Ecological Remediation for Lake & River, Hubei University of Technology, Wuhan 430068, China; 15327367382@163.com (M.G.); lk52598@163.com (K.L.); m18803988359@163.com (H.W.); songyilin0610@163.com (Y.S.); lyy818927@163.com (Y.L.); 13361590405@163.com (W.Z.); jgao13@hotmail.com (J.G.); 2Key Laboratory of Intelligent Health Perception and Ecological Restoration of Rivers and Lakes, Ministry of Education, Hubei University of Technology, Wuhan 430068, China

**Keywords:** halophilic denitrifying bacteria, saline wastewater, engineering application

## Abstract

To date, the nitrogen metabolism pathways and salt-tolerance mechanisms of halophilic denitrifying bacteria have not been fully studied, and full-scale engineering trials with saline fly ash-washing wastewater have not been reported. In this study, we isolated and screened a halophilic denitrifying bacterium (*Marinobacter* sp.), GH-1, analyzed its nitrogen metabolism pathways and salt-tolerance mechanisms using whole-genome data, and explored its nitrogen removal characteristics under both aerobic and anaerobic conditions at different salinity levels. GH-1 was then applied in a full-scale engineering project to treat saline fly ash-washing leachate. The main results were as follows: (1) Based on the integration of whole-genome data, it is preliminarily hypothesized that the strain possesses complete nitrogen metabolism pathways, including denitrification, a dissimilatory nitrate reduction to ammonium (DNRA), and ammonium assimilation, as well as the following three synergistic strategies through which to counter hyperosmotic stress: inorganic ion homeostasis, organic osmolyte accumulation, and structural adaptations. (2) The strain demonstrated effective nitrogen removal under aerobic, anaerobic, and saline conditions (3–9%). (3) When applied in a full-scale engineering system treating saline fly ash-washing wastewater, it improved nitrate nitrogen (NO_3_^−^-N), total nitrogen (TN), and chemical oxygen demand (COD) removal efficiencies by 31.92%, 25.19%, and 31.8%, respectively. The proportion of *Marinobacter* sp. increased from 0.73% to 3.41% (aerobic stage) and 2.86% (anoxic stage). Overall, halophilic denitrifying bacterium GH-1 can significantly enhance the nitrogen removal efficiency of saline wastewater systems, providing crucial guidance for biological nitrogen removal treatment.

## 1. Introduction

With the increasing proportion of municipal solid waste (MSW) incineration in global waste management, fly ash production has become a critical environmental challenge. In some European countries, such as Finland, the Netherlands, and Denmark, approximately 50% of total MSW is treated via incineration [1]. In China, incineration accounted for 79.86% of harmless MSW disposal in 2021, annually generating over 18 million tons of fly ash [2]. Fly ash, constituting 3–15% of incineration residues, contains high concentrations of chlorides, sulfates, and heavy metals (e.g., Cr, Hg), posing significant risks to ecosystems and human health [3].

Water-washing has emerged as a widely adopted pretreatment method to reduce chloride content and enhance fly ash stability for subsequent recycling or landfilling [4]. However, leachate from fly ash-washing retains elevated nitrate (NO_3_^−^), as well as residual chloride (Cl^−^), sulfate (SO_4_^2−^), and other bio-toxic compounds [5]. Notably, the complex composition of such wastewater leads to osmotic stress and suppresses microbial activity, resulting in inefficient denitrification and system instability [6,7].

Recent studies have identified halotolerant denitrifying bacteria as potential candidates for addressing high-salinity wastewater. Strains such as *Acinetobacter junii* YB, *Vibrio diabolicus* SF16, and *Vibrio* sp. exhibit the dual capabilities of salt tolerance and efficient nitrate removal [8,9,10]. Nevertheless, most research remains confined to laboratory-scale experiments, with limited validation in full-scale applications [8,11]. Furthermore, the metabolic pathways and salt-adaptation mechanisms of these strains are poorly characterized, hindering their industrial scalability [12].

In order to bridge this gap, we isolated a novel *Marinobacter* sp. strain from high-salinity tannery wastewater sludge for this study. The strain’s denitrification performance under variable dissolved oxygen (DO) and salinity levels was systematically investigated; this process was complemented by whole-genome analysis to elucidate its nitrogen metabolism pathways and haloadaptation mechanisms. Full-scale engineering trials were conducted to validate its efficacy in treating fly ash-washing leachate, providing a framework for the industrial application of halophilic denitrifiers.

## 2. Materials and Methods

### 2.1. Strain Screening and Isolation

Activated sludge samples were collected from the aerobic tank of a tannery’s wastewater treatment system. A 5 mL aliquot of sludge was inoculated into 100 mL of DM (Aerobic Denitrification Medium) and incubated at 30 °C with shaking at 150 rpm for 7 days. Subsequently, 5 mL of the enriched culture was transferred to fresh DM and incubated under identical conditions for an additional 7 days; this serial enrichment was repeated for five consecutive cycles. The enriched culture was then serially diluted (10^−2^ to 10^−9^) and 0.1 mL aliquots of each dilution were spread onto DM agar plates. After incubation at 30 °C until visible colonies formed, single colonies were purified through five successive streaking procedures.

The DM, modified from a previous formula [10], consisted of the following components per liter of distilled water: 1.45 g of KNO_3_, 6.83 g of CH_3_COONa, 0.10 g of MgSO_4_·7H_2_O, 7.90 g of K_2_HPO_4_·3H_2_O, 1.50 g of KH_2_PO_4_, 50 g of NaCl, and 2 mL of trace elements. The trace element solution contained the following (per liter of distilled water): 63.70 g of Na_2_EDTA, 5.50 g of CaCl_2_, 3.90 g of ZnSO_4_·7H_2_O, 5.06 g of MnCl_2_·4H_2_O, 5.00 g of FeSO_4_·7H_2_O, 1.00 g of Na_2_MoO_4_·2H_2_O, 1.01 g of CuSO_4_, and 1.61 g of CoCl_2_·6H_2_O.

### 2.2. Strain Morphology and Species Identification

Approximately 1.5 g of the purified microbial sample was selected and 2.5% glutaraldehyde was used as the chemical fixative. The sample was immersed for 30 min for fixation, followed by washing three times with 0.1 M phosphate-buffer solution, for ten minutes each time; after washing, the sample was dehydrated using a gradient concentration of ethanol (30%, 50%, 70%, and 90%), and then subjected to freeze-drying in a freeze-dryer. The dried sample was treated for conductivity and observed under a scanning electron microscope (SEM) at a magnification of 18,000 times to examine the morphology of the strain. The SEM model used was the ZEISS Sigma 300.

The taxonomic identification of purified strains was conducted, with Genomic DNA from the purified isolates amplified using 27F (5′-AGAGTTTGATCCTGGCTCAG-3′) and 1492R (5′-GGTTACCTTGTTACGACTT-3′) primers, targeting the 16S rDNA gene [13]. The PCR procedure included the following steps: initial denaturation at 95 °C for 5 min, followed by 35 cycles of 95 °C for 40 s, 55 °C for 40 s, and 72 °C for 40 s, with a final extension of 72 °C for 10 min. The PCR products were sequenced using Sangon Biotech (Shanghai, China). The resulting 16S rDNA sequences were aligned with the NCBI GenBank database using BLAST (version 2.16.0) (http://www.ncbi.nlm.nih.gov/BLAST/Blast.cgi, accessed on 27 May 2024), and phylogenetic trees were constructed using the neighbor-joining method in MEGA 11.056.

### 2.3. Genomic Sequencing

The purpose of the whole-genome sequencing of the purified strains was to explain and predict the nitrogen metabolism pathways and halotolerance mechanisms of the strain at the genetic level. Genomic DNA was extracted using the SDS method and sequenced using Majorbio (Shanghai, China). The sequencing approach combined Illumina and PacBio technologies. First, purified genomic DNA was collected and fragmented using Covaris, a technology that employs high-frequency sound waves (ultrasound) to precisely shear DNA molecules into specific-sized fragments. The ends of the DNA fragments were repaired to create blunt ends, facilitating the subsequent ligation of sequencing adapters. Sequencing adapters were then ligated to both ends of the DNA fragments. Fragments of the desired size were selected through gel electrophoresis or magnetic bead purification, and unligated adapters and impurities were removed. The adapter-ligated DNA fragments were amplified using PCR to increase their concentration in the library, thereby constructing the genomic sequencing library. After library preparation, sequencing was performed using both the Illumina and the PacBio platforms. The raw reads generated from sequencing underwent quality control, alignment, variant detection, and annotation analyses to ultimately obtain whole-genome sequence information. Coding genes were functionally annotated using the Kyoto Encyclopedia of Genes and Genomes (KEGG) database, and metabolic pathways were reconstructed using the KEGG Mapper. The sequencing data were deposited in the NCBI (National Center for Biotechnology Information) Sequence Read Archive database under the accession number PRJNA1228538.

### 2.4. Denitrification Characteristics of the Strain

To evaluate the denitrification characteristics of the strain under aerobic and anaerobic conditions, the purified strain, cultured to the logarithmic growth phase (OD600 ≈ 1), was inoculated at a 1% (*v*/*v*) ratio into 150 mL of DM and incubated under aerobic and anaerobic conditions. Under aerobic conditions, the conical flasks were sealed with breathable sealing film and placed in a constant-temperature shaker at 30 °C and 150 rpm (dissolved oxygen, DO = 6.8 mg/L). For anaerobic conditions, 150 mL of DM was purged with sterile nitrogen for 5 min, followed by the addition of a 1 cm layer of sterile liquid paraffin on the surface of the medium (DO < 0.1 mg/L); the flasks were then sealed with an airtight plastic film and statically incubated at 30 °C. Each experiment was performed in triplicate, and samples were collected every 6 to 12 h to measure OD600 values, as well as the concentrations of ammonia nitrogen (NH_4_^+^-N), nitrite nitrogen (NO_2_^−^-N), nitrate nitrogen (NO_3_^−^-N), and total nitrogen (TN). The detection methods were obtained from the Standard Methods for the Examination of Water and Wastewater [14]. The OD600 value of the sample was detected using a multifunctional microporous plate (Lx, BioTek, Winooski, VT, USA). The following methods were used: the salicylic acid–hypochlorous acid method for measuring NH_4_^+^-N, the N-(1-naphthyl)-ethylenediamine dihydrochloride method (NED method) for measuring NO_2_^−^-N, the aminosulfonic acid method for measuring NO_3_^−^-N, and the alkaline potassium persulfate method for measuring TN.

To assess the denitrification characteristics of the strain under different salinity conditions, the purified strain cultured to the logarithmic growth phase (OD600 ≈ 1) was inoculated at a 1% (*v*/*v*) ratio into 150 mL of DM with varying salinity gradients. Six salinity gradients (0%, 3%, 5%, 7%, 9%, and 12%) were established by adjusting NaCl concentrations in the DM, with triplicate setups for each gradient. After inoculation, the conical flasks were sealed with breathable sealing film and placed in a constant-temperature shaker at 30 °C and 150 rpm. Samples were collected every 6 to 12 h to measure OD600 values, as well as the concentrations of NH_4_^+^-N, NO_2_^−^-N, NO_3_^−^-N, and TN. The detection methods were the same as those described above.

### 2.5. Application in Full-Scale Fly Ash Leachate Treatment System

The experimental facility is located in Zhejiang Province, China, and employs a water–acid-washing treatment process for the harmless utilization of fly ash derived from municipal solid waste (MSW) incineration. Approximately 1200 m^3^ of washing water for fly ash is generated daily. After heavy metal removal, the water is mixed with small amounts of other wastewaters, collectively referred to as raw water (the composition of which is detailed in the Appendix A). The raw water is then fed into an A/O (anoxic/oxic) treatment system, and, after meeting the discharge standards, it is discharged into the municipal sewage network. The A/O treatment system is a widely used technology in wastewater treatment, effectively removing organic matter, nitrogen, phosphorus, and other pollutants through the combination of anoxic and oxic stages. A process flow diagram of the A/O treatment system is provided in the Appendix A. The anaerobic tank of the A/O treatment system has a volume of 1216 m^3^, the aerobic tank has a volume of 680 m^3^, the hydraulic retention time is 2 h in the anaerobic tank and 6 to 8 h in the aerobic tank, the mixed liquor recirculation ratio is 400%, and the sludge recirculation ratio is 80%. The water quality characteristics include a nitrate nitrogen concentration ranging from 150 mg/L to 350 mg/L, an ammonia nitrogen content below 20 mg/L, and a salinity level between 2% and 4%. A mixed carbon source (primarily composed of small molecular acids, sugars, alcohols, etc.) is supplemented into the anaerobic tank to maintain a C/N ratio of 4 to 6.

Five liters of pure strain cultures, grown under laboratory conditions with a bacterial concentration of 8 × 10^9^ to 2 × 10^10^ CFU/mL, were inoculated into a 500 L tank for primary expansion cultivation. Once cultivated to the log phase, the cultures were pumped into a secondary expansion cultivation tank with a volume of 5.6 m^3^. Raw water, supplemented with a carbon source, was used as the expansion cultivation medium; the aeration rate was controlled to maintain dissolved oxygen levels between 2 and 4 mg/L, and stirring devices were continuously operated. After 3 to 5 days of cultivation in the secondary expansion tank, the cultivated halophilic denitrifying activated sludge was inoculated into the aerobic tank of the mainstream system (the volume of the aerobic tank is 680 m^3^). Five batches of halophilic denitrifying activated sludge were cultivated and inoculated into the aerobic tank on Days 17, 27, 38, 43, and 55. For Batch 2, inoculation was performed in three stages (Days 27, 29, and 31), whereas all other batches were inoculated in a single operation on their designated days. The experiment lasted for 60 days and was divided into six stages based on the timing and amount of inoculum added, as detailed in Table 1. Water quality levels in the anaerobic tank inflow and aerobic tank outflow were monitored every 24 h throughout the experimental period.

### 2.6. Analysis of the Microbial Community Structure

To evaluate the effectiveness of bioaugmentation, the changes in the microbial community structure of the biochemical treatment system before and after the addition of microbial agents were analyzed. Activated sludge samples (50 g each) from the aerobic and anaerobic tanks were collected before and after the addition, stored at −80 °C, and sent to Majorbio (Shanghai, China) for analysis. Genomic DNA was extracted using the E.Z.N.A.^®^ Soil DNA Kit (Omega Bio-tek, Norcross, GA, USA). The V3-V4 region of the 16S rRNA gene was amplified with PCR using the forward primer 338F (5′-ACTCCTACGGGAGGCAGCAG-3′) and the reverse primer 806R (5′-GGACTACHVGGGTWTCTAAT-3′). The PCR products were detected through 2% agarose gel electrophoresis and recovered using the AxyPrep DNA Gel Extraction Kit (AXYGEN). The recovered PCR products were purified using the DNA Gel Recovery and Purification Kit (PCR Clean-Up Kit, China Yuhua). The purified PCR products were then used for library construction with the TruSeqTM DNA Sample Prep Kit and sequenced on the Illumina Nextseq2000 platform. All sequences were clustered into Operational Taxonomic Units (OTUs) [15] at a 97% similarity threshold for bioinformatics analysis. Taxonomic annotation of the OTUs was performed, and the community composition of each sample was statistically analyzed at different taxonomic levels with a confidence threshold of 70% [16].

### 2.7. Data Processing Methods

Inter-group differences in water quality data were analyzed using one-way ANOVA with SPSS 20. MEGA 10.1.7 was utilized to construct phylogenetic trees (https://www.megasoftware.net/, accessed on 27 May 2024) and SOAPdenovo 2.04 was used to assemble sequencing data, while the KEGG Database 202209 was employed to annotate strain-related metabolic pathways (http://www.genome.jp/kegg/, accessed on 18 March 2025). Origin 9 was utilized for data visualization.

The nitrate removal rate (NRR) and total nitrogen removal efficiency (TNRE) were calculated using the following methods [17,18].

The nitrate removal rate was calculated using Equation (1), as follows:(1)$$NRR\left(mg/(L\cdot h)\right)=\frac{C_{1}-C_{2}}{t_{1}-t_{2}}$$
where $C_{1}$ is the corresponding concentration of nitrate at time $t_{1}$, in mg/L, and $C_{2}\;$ is the corresponding concentration of nitrate at time $t_{2}$, in mg/L, with $t_{1}$ and $t_{2}$ representing distinct time points, in hours.

The total nitrogen removal efficiency is calculated using Equation (2), as follows:(2)$$TNRE\;\left(\%\right)=\frac{{TN}_{initial}-{TN}_{final}}{{TN}_{initial}}\times 100\%$$
where ${TN}_{initial}$ is the concentration of total nitrogen at the commencement of the experiment, and ${TN}_{final}$ is the concentration of total nitrogen at the conclusion of the experiment, both measured in mg/L.

## 3. Results

### 3.1. Strain Identification

The strain exhibits a rod-shaped morphology, with lengths ranging from 1.0 to 1.5 μm (Figure 1); it is Gram-negative and non-sporulating, and it lacks a capsule. On a solid medium, single colonies appear opaque, faintly yellow, smooth, and circular, with well-defined edges. The strain showed 100% similarity to *Marinobacter shenglinensis* in a 16S rDNA homology comparison. The phylogenetic tree is presented in the Appendix A. We named this strain GH-1 (GenBank accession number: PP853485).

### 3.2. Whole-Genome Information and Functional Gene Annotation

Whole-genome sequencing revealed that strain GH-1 possesses a circular chromosome spanning 3.95 Mb, with a GC content of 57.88% and an average protein-coding gene length of 1004.81 bp (Figure 2a). The SRA accession number for the whole-genome data is PRJNA1228538. In total, 3934 protein-coding genes were annotated, including key nitrogen metabolism-related genes encoding nitrate reductases (napAB/narGHI), nitrite reductases (nirS/nirBD), nitric oxide reductase (norBC), nitrous oxide reductase (nosZ), glutamine synthetase (glnA), and glutamate synthase (gltB) (Figure 2b). No nitrification-associated genes (amo, hao) were identified.

Through a comparative analysis of the GH-1 genome and the NR database, genes related to sodium and potassium metabolism were annotated, including nhaB (Na^+^/H^+^ antiporter), kdpA/B/C/D/E (high-affinity ATP-driven K^+^ transport system), and trkA/H (potassium transporter). Additionally, the following functionally relevant genes were annotated: betA/B (betaine–aldehyde dehydrogenase), ectA/B/C/D (ectoine synthase), proA/B/C (proline synthesis), cfa (cyclopropane fatty acid synthase), phoD (alkaline phosphatase), and maeB (malate dehydrogenase). For detailed information, please refer to the Appendix A.

### 3.3. Oxygen and Salinity Effects on Nitrogen Removal Efficiency of Strain GH-1

#### 3.3.1. Oxygen Effects

As shown in Figure 3, strain GH-1 acquired a maximum OD600 of 0.65 under aerobic conditions, representing a 9.29-fold increase compared to that obtained under anaerobic conditions (0.07). Complete nitrate consumption occurred under both conditions, with the maximum aerobic nitrate removal rate (11.79 ± 0.37 mg/(L·h)) surpassing the anaerobic rate (6.81 ± 0.85 mg/(L·h)). However, the total nitrogen removal efficiency was higher under anaerobic conditions (88.95 ± 0.47%) than under aerobic conditions (68.15 ± 1.18%). Maximum nitrite accumulation levels remained comparable between the environments (1.15 ± 0.09 mg/L under aerobic vs. 1.04 ± 0.07 mg/L under anaerobic conditions).

#### 3.3.2. Salinity Effects

Figure 4a reveals extended lag phases for strain GH-1 with increasing salinity (16 h at 3% vs. 88 h at 9%). No growth occurred at 0% or 12% salinity. Experimental studies indicated that halophilic bacterial growth exhibited a strict dependence on ionic strength, with complete inhibition observed under both non-saline (0% NaCl) and hypersaline conditions. Nitrate removal efficiency consistently exceeded 97.50%, with no statistically significant differences (*p* > 0.05) across the salinity range of 3–9% (Figure 4b). In contrast, total nitrogen removal efficiency gradually declined, from 77.13 ± 6.08% at 3% salinity to 58.28 ± 4.10% at 9% salinity (Figure 4d). Concurrently, maximum nitrite accumulation increased from 15.66 ± 0.81 mg/L to 43.89 ± 3.70 mg/L over the same salinity gradient (Figure 4c).

### 3.4. Bioaugmentation Effects of Strain GH-1 at Full Scale

#### 3.4.1. Enhancement of Treatment Efficiency

Figure 5 illustrates the changes in water quality before and after bioaugmentation. The entire operation cycle was divided into six stages based on the batches of microbial inoculants added. During the operation stage without inoculants (the first 17 days), the influent concentrations of NO_3_^−^-N, TN, COD, and NH_4_^+^-N were maintained at 143.00–265.14 mg/L, 173.86–305.96 mg/L, 940.37–1513.15 mg/L, and 7.70–15.70 mg/L, respectively, with corresponding average removal efficiencies of 60.08%, 57.81%, 63.21%, and 19.70%. During this stage, the overall efficiency of nitrogen and organic matter removal was poor, and the effluent water quality failed to meet discharge standards, particularly for TN and COD.

After adding the inoculants, the system performance significantly improved, achieving stable and compliant effluent quality. In the first stage after inoculant addition (days 18–27), despite a significant increase in influent pollutant concentrations (NO_3_^−^: 216–386 mg/L; TN: 285–422 mg/L; COD: 1531–2126 mg/L), the average removal efficiencies increased to 76.20%, 70.38%, and 78.93%, respectively. In the second stage after inoculant addition, despite large fluctuations in influent water quality, the treatment efficiency further improved, with the average removal rates for NO_3_^−^, TN, and COD increasing to 85.70%, 78.30%, and 86.25%, respectively. From the third to the fifth stages, the system stabilized, with the NO_3_^−^, TN, and COD removal efficiencies reaching 92.45%, 83.08%, and 92.54%, respectively. By the end of the experiment, compared to pre-inoculation levels, the NO_3_^−^, TN, and COD removal efficiencies increased by 31.92%, 25.19%, and 38.19%, respectively, while the ammonia nitrogen removal efficiency consistently remained above 20.12%.

#### 3.4.2. Impact of Bioaugmentation on Microbial Community Structure

In the secondary expansion cultivation tank, *Proteobacteria* dominated, with an overwhelming proportion of 98.22% (Figure 6a). Through bioaugmentation, the proportion of *Proteobacteria* in the mainstream system also significantly increased, rising from 67.11% to 85.84% in the aerobic section and 89.30% in the anoxic section. Simultaneously, the proportions of *Chloroflexi*, *Actinobacteria*, *Bacteroidota*, and *Firmicutes* all decreased.

Figure 6b shows the changes in the relative abundance of bacterial genera. Compared to before bioaugmentation, the proportion of *Thauera* significantly increased in the mainstream system afterward, rising from 13.49% to 47.73% in the aerobic section and 53.88% in the anoxic section, becoming the dominant genus. Although *Marinobacter* accounted for 46.27% of the secondary enrichment culture system, indicating an ability to adapt to the water quality characteristics of the fly ash-washing wastewater and achieve rapid proliferation, in the mainstream system, the proportion of *Marinobacter* only increased from 0.73% to 3.41% in the aerobic section and 2.86% in the anoxic section, making it the fourth most dominant species in the system.

## 4. Discussion

### 4.1. Nitrogen Metabolic Pathways and Halophilic Mechanisms of Strain GH-1

It has been demonstrated in previous studies that a 16S rRNA sequence similarity greater than 97% is indicative of two microorganisms belonging to the same genus [19]. Based on the 16S rDNA homology comparison, strain GH-1 shows 100% similarity to *Marinobacter shengliensis*. To date, seven complete genome sequences of *Marinobacter shengliensis* have been reported. While Yasuda and Terada documented the genome architecture of *M. shengliensis* D49, they omitted nitrogen metabolism gene annotations [20]. Subsequent studies by Elkassas et al. only identified dissimilatory nitrate reductase and nitrite reductase genes in six other *M. shengliensis* strains, and no nitrogen metabolic pathway was analyzed [21]. The nitrogen metabolism pathway of strain GH-1 is similar to that of most studied strains. For instance, GH-1 and the aerobic denitrifying bacterium JI-2 possess analogous denitrification protease genes and share the same denitrification pathway, as follows: NO_3_^−^-N → NO_2_^−^-N → NO → N_2_O → N_2_ [22]. napAB, a periplasmic protein, and narGHI, a membrane-bound protein, both catalyze the reduction of nitrate to nitrite [23]. Nitrite reductase (nirS), nitric oxide reductase (norBC), and nitrous oxide reductase (nosZ) participate in the subsequent steps of denitrification, sequentially reducing nitrite to nitrogen gas [24,25,26]. In the DNRA (dissimilatory nitrate reduction to ammonium) process, nitrite reductase (nirBD) catalyzes the reduction of nitrite to ammonium [27]. During ammonium assimilation, intracellular ammonium is converted to L-glutamine via glutamine synthetase (glnA) and further transformed into L-glutamate by glutamate synthase (gltB) [28]. Finally, L-glutamate enters the glutamate metabolism, participating in protein synthesis or being converted into glucose, among other compounds. Based on whole-genome data, it is preliminarily hypothesized that strain GH-1 possesses the following three primary nitrogen metabolism pathways: denitrification, DNRA, and ammonium assimilation (Figure 7). With this study, we provide the first comprehensive nitrogen metabolic network reconstruction for this species, revealing the reasons for its genetic absence in previous reports.

Based on the whole-genome data, it is preliminarily hypothesized that strain GH-1 employs the following three synergistic strategies to combat hyperosmotic stress: inorganic ion homeostasis, organic osmolyte accumulation, and structural adaptations (Table 2). The nhaB-mediated Na^+^ extrusion system coordinates with kdp/trk-dependent K^+^ uptake to maintain a cytoplasmic Na^+^/K^+^ balance under hypersaline conditions [29,30]; this dual-transport mechanism prevents cellular dehydration through ionic osmoregulation [31]. The concurrent expressions of betA/B, ectA/B/C/D, and proA/B/C facilitate the biosynthesis of betaine, ectoine, and proline, key compatible solutes that stabilize macromolecular structures without disrupting enzymatic activity [32]; these zwitterionic compounds exhibit a water retention capacity proportional to the external NaCl concentration [29]. Cyclopropane fatty acid synthase, encoded by the gene cfa, modifies unsaturated fatty acids through cyclopropane ring formation, enhancing membrane rigidity and reducing lipid bilayer permeability under osmotic shock [33]. Halophilic phoD and maeB maintain catalytic efficiency at an elevated ionic strength using surface-exposed acidic residues that coordinate hydration shells [34]; this genomic architecture aligns with the “salt-in-cytoplasm” strategy observed in moderately halophilic γ-proteobacteria [35]. While our findings establish genomic prerequisites for halotolerance, transcriptomic profiling under salt gradients and the metabolomic validation of ectoine/betaine accumulation patterns remain essential for functional confirmation.

### 4.2. Impacts of Different Culture Conditions on Nitrogen Removal Efficiency of Strain GH-1

Compared to anaerobic conditions, aerobic conditions exhibit higher biomass and nitrate removal rates (Figure 3), as bacteria utilize oxygen as the terminal electron-acceptor in aerobic environments, generating more energy [40] and thereby promoting bacterial growth and resulting in an increased biomass and increased growth rates. The total nitrogen removal efficiency of strain GH-1 under anaerobic conditions was 20.80% higher than that under aerobic conditions, which is attributed to the fact that a high level of dissolved oxygen affects the synthesis and activity of microbial denitrification enzymes (such as Nar, Nir, Nor, and Nos) [41], while also facilitating the DNRA process [42], thereby reducing the total nitrogen removal efficiency. Under aerobic conditions, strain GH-1 converted 24.74 ± 1.12 mg/L of total nitrogen into biomass nitrogen, which typically exists in the form of activated sludge in practical engineering applications and is eventually discharged from the system; therefore, in such applications, the total nitrogen removal efficiency of strain GH-1 under aerobic conditions is not lower than that under anaerobic conditions, and the removal rate is higher.

Experimental studies indicate that halophilic bacterial growth exhibited strict dependence on ionic strength, with complete inhibition observed under both non-saline (0% NaCl) and hypersaline conditions (12% NaCl) (Figure 4), a phenomenon that primarily stems from the salt-dependent conformational stability of halophilic enzymes, which require optimal ion concentrations to maintain catalytic functionality through specific residue–ion interactions [43,44]. Furthermore, hypertonic environments that exceed the cellular osmoregulation capacities induce structural damage to membrane systems and cytoplasmic components, ultimately leading to the irreversible loss of cellular viability [38,39]. The salinity-dependent lag-phase extension reflects the time requirements for osmotic adaptation through compatible solute synthesis [45]. Within the salinity range of 3% to 9%, the total nitrogen removal rate gradually decreased, while the maximum accumulation of nitrite progressively increased, observations which suggest that elevated salinity triggers a systematic redistribution of nitrate metabolic flux. Specifically, assimilatory nitrogen pathways are progressively enhanced, while denitrification activity is simultaneously suppressed, a metabolic shift which likely stems from salinity-induced reductions in both enzymatic activity and the abundance of key denitrifying enzymes (e.g., periplasmic Nar, Nir, and Nos) [46,47]; this trend is further corroborated by the observed accumulation of nitrite nitrogen under elevated-salinity conditions. The metabolic redistribution may also represent a microbial survival strategy involving the intracellular accumulation of amino acids, purines, and other osmolytes to counterbalance osmotic stress in high-salinity environments, which subsequently enhances nitrogen assimilation efficiency through the redirection of metabolic flux [48,49]. Intriguingly, no significant difference in total nitrogen removal efficiency was observed within the salinity range of 3–5%, indicating the strain’s tolerance to moderate salinity shocks and its capacity to maintain stable denitrification performance under such conditions.

### 4.3. Bioaugmentation of Strain GH-1 in Full-Scale Systems and Microbial Community Dynamics

Previous studies have shown that the inoculation of halophilic denitrifying bacteria under pilot-scale conditions can significantly improve the removal efficiency of nitrogen and carbon in high-salinity wastewater treatment systems. For instance, Gao [8] applied salt-tolerant *Vibrio* sp. to a moving-bed bioreactor (MBBR), achieving an average increase of 26.74% in nitrogen removal efficiency and 10.15% in carbon removal efficiency. Similarly, Fu [11] introduced salt-tolerant aerobic denitrifying bacteria into constructed wetlands, attaining removal efficiencies of 79.2% for NH_4_^+^-N, 95.7% for NO_3_^−^-N, and 89.9% for TN in saline aquaculture wastewater. However, studies on the full-scale bioaugmentation of salt-tolerant bacteria in high-salinity wastewater treatment are still scarce, and their efficacy is frequently undermined by fluctuations in operational parameters, such as temperature, pH, dissolved oxygen, and pollutant loading variability, leading to disruptions in microbial metabolic activity and reductions in nitrogen removal efficiency [50]. Furthermore, competition with indigenous microbial communities and predation pressure frequently hinder the colonization and functional dominance of introduced strains [51,52]. Remarkably, strain GH-1 exhibited a significant enhancement in nitrogen removal efficiency in full-scale wastewater treatment systems, demonstrating its strong environmental adaptability. While the TN removal efficiency demonstrated a progressive enhancement, a concomitant accumulation of NH_4_^+^-N was detected, a phenomenon that could potentially be attributed to the suppression of glutamate metabolism pathways. However, further transcriptomic analyses are required to validate this hypothesis.

*Proteobacteria* is the most common and largest phylum in wastewater treatment, containing a large number of nitrogen-removing bacteria [53]. The decrease in *Chloroflexi*, *Actinobacteria*, *Bacteroidota*, and *Firmicutes* may have been due to the competitive advantage of *Proteobacteria*. *Marinobacter* increased from 0.73% to 3.41% in the aerobic section and 2.86% in the anoxic section, making it the fourth most dominant species. However, *Thauera* became the dominant genus in the mainstream system. *Thauera* plays an important role in various wastewater treatment systems, effectively removing nitrate and COD from wastewater [54,55,56]. During the process of exogenous bacterial bioaugmentation, the introduced bacteria may fail to become dominant species due to competition with indigenous microorganisms [57,58]. At the same time, many studies have found that, even though exogenous strains cannot survive or maintain dominance in a system, the performance of the biological treatment system still improves [59,60,61]. Some studies suggest that the bioaugmentation effect of exogenous strains may be achieved through mechanisms such as changing the microbial community’s structure, inducing the formation of dominant degradative bacteria, and selectively enriching specific functional bacteria, ultimately achieving the effective metabolism of target pollutants [62]. Therefore, we speculate that strain GH-1, while performing its own nitrogen removal function, may also regulate the microbial community’s structure and promote the growth of *Thauera*, further enhancing the system’s nitrogen removal capacity.

## 5. Conclusions

In this study, we isolated a halophilic denitrifying bacterium, GH-1 (*Marinobacter* sp.), from high-salinity tannery wastewater sludge. Based on whole-genome data analysis, it is preliminarily hypothesized that strain GH-1 possesses three nitrogen metabolic pathways: denitrification, DNRA, and ammonium assimilation. Additionally, it adapts to external osmotic pressure changes through inorganic ion homeostasis, the accumulation of organic osmolytes, and structural adaptations. The strain exhibited excellent nitrogen removal characteristics under aerobic and anaerobic conditions, as well as saline concentrations ranging from 3% to 9%. When applied to a full-scale project treating saline fly ash-washing wastewater, the NO_3_^−^N, TN, and COD removal rates increased by 31.92%, 25.19%, and 31.8%, respectively. The proportion of *Marinobacter* sp. rose from 0.73% to 3.41% (aerobic stage) and 2.86% (anoxic stage), making it the fourth most dominant species in the system. In this study, we elucidated the salt-tolerance mechanisms of *Marinobacter* sp., opened new perspectives for the engineering applications of halophilic bacteria in treating saline wastewater, and laid the foundation for subsequent research. Future research could further explore whether the *Marinobacter* sp. strain possesses the potential to reduce N_2_O emissions through complete denitrification, aiming to increase its application value.

## Figures and Tables

**Figure 1 microorganisms-13-01274-f001:**
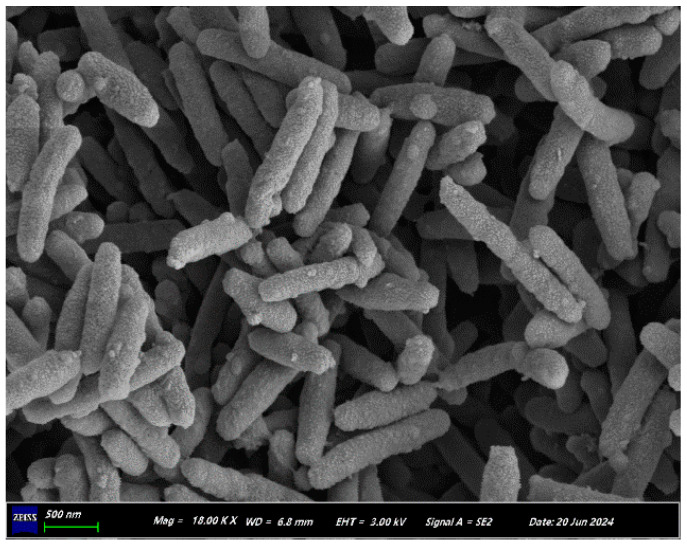
The morphology of strain GH-1 observed under a scanning electron microscope.

**Figure 2 microorganisms-13-01274-f002:**
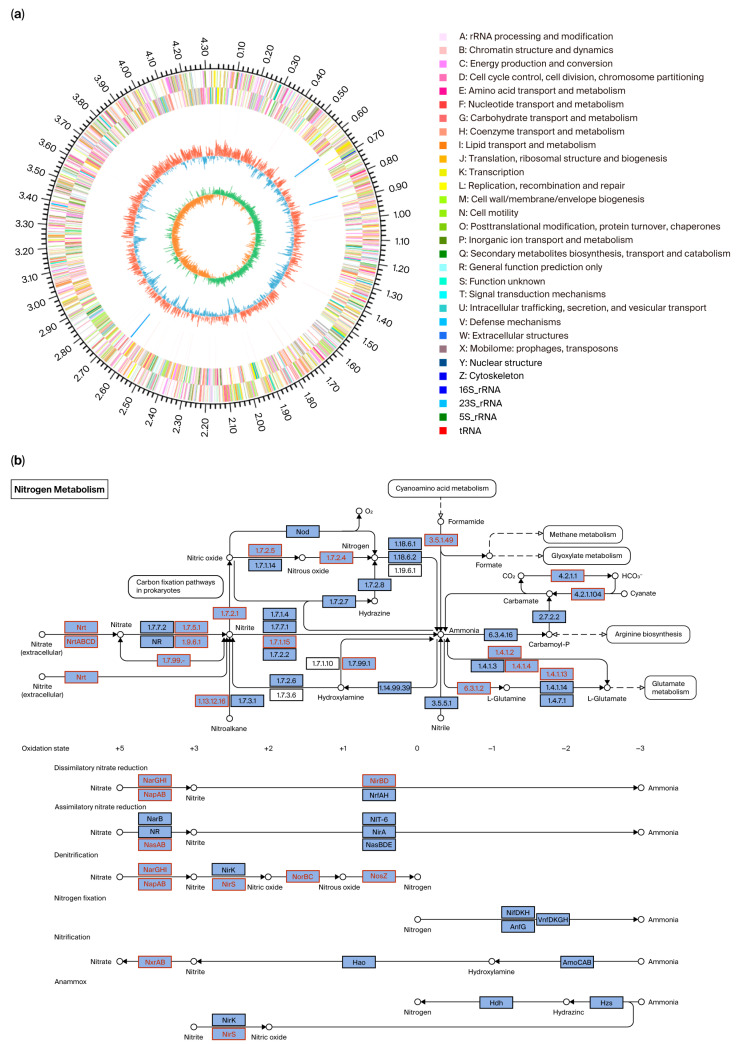
Whole-genome sequencing information for strain GH-1. (**a**) A Circos diagram of the entire genome. (**b**) A predicted nitrogen metabolism pathway diagram of strain GH-1.

**Figure 3 microorganisms-13-01274-f003:**
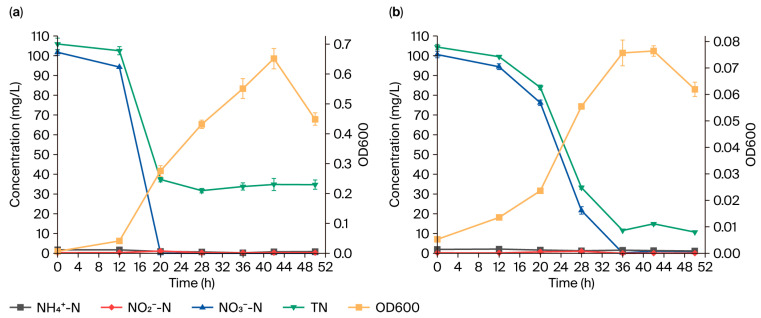
Nitrogen removal efficiency of strain GH-1 under (**a**) aerobic and (**b**) anaerobic conditions.

**Figure 4 microorganisms-13-01274-f004:**
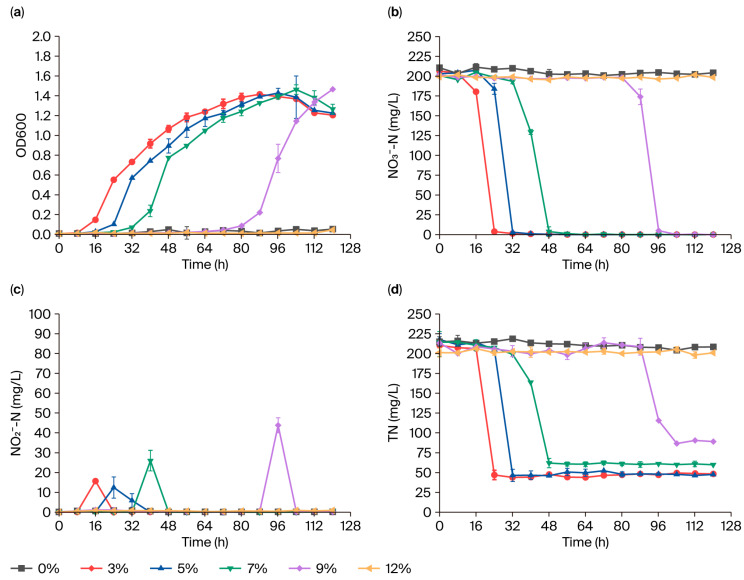
Nitrogen removal efficiency of strain GH-1 under different salinity conditions: (**a**) OD600, (**b**) NO_3_^−^-N, (**c**) NO_2_^−^-N, and (**d**) TN.

**Figure 5 microorganisms-13-01274-f005:**
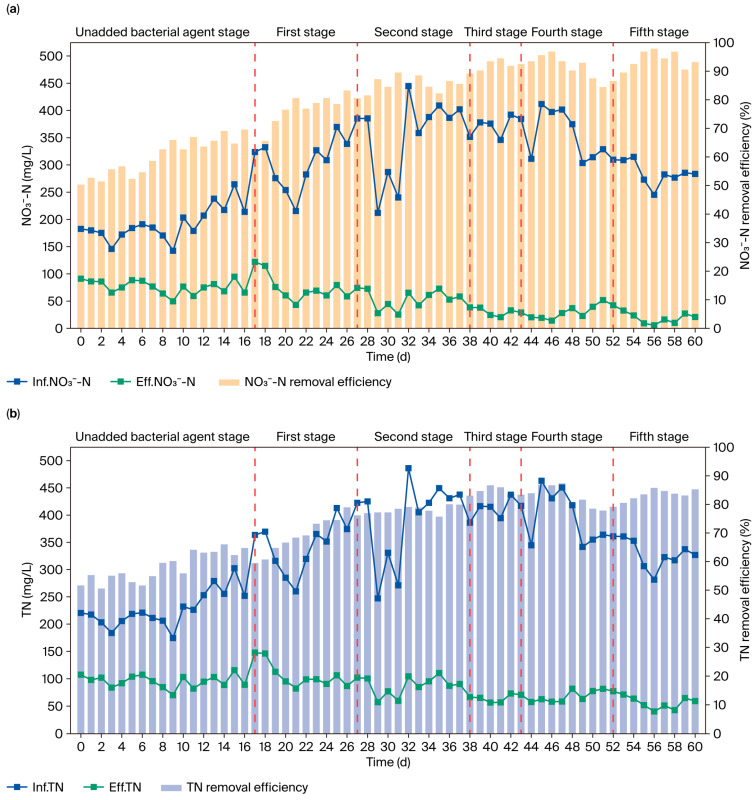
Changes in water quality before and after the addition of microbial agents: (**a**) NO_3_^−^-N, (**b**) TN, (**c**) COD, and (**d**) NH_4_^+^-N concentrations.

**Figure 6 microorganisms-13-01274-f006:**
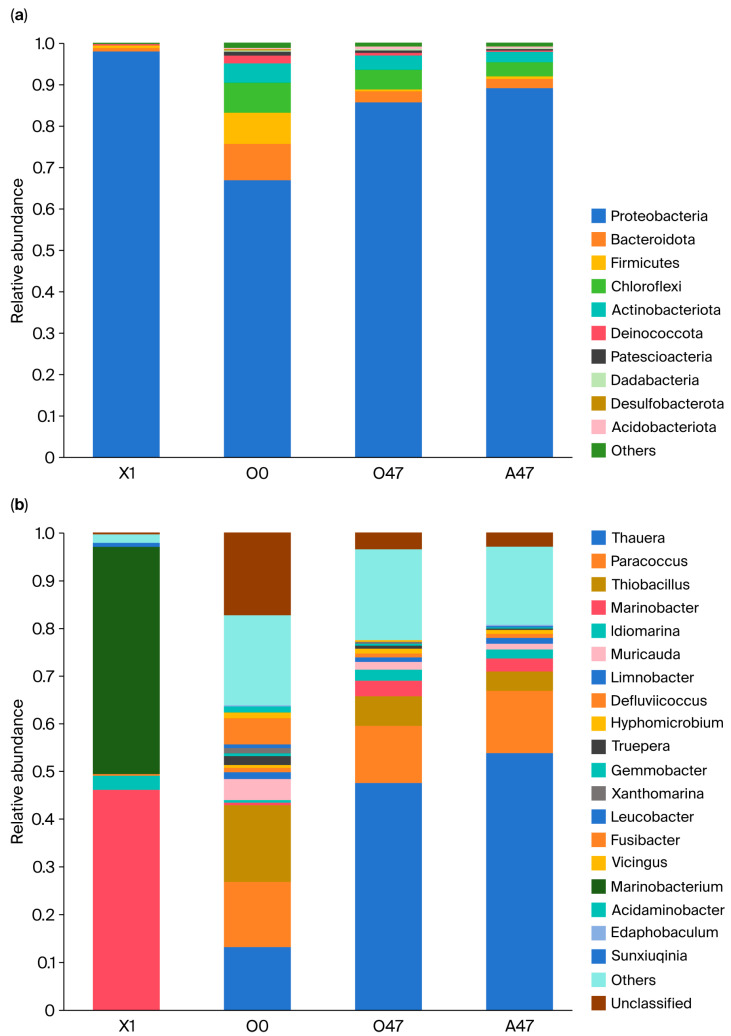
The structure of microbial communities at the phylum and genus levels. (**a**) The relative abundance of microbial communities at the phylum level. (**b**) The relative abundance of microbial communities at the genus level. X1: A microbial sample from the secondary expansion tank; O0: A microbial sample from the aerobic tank on day 0 of system operation; O47: A microbial sample from the aerobic tank on day 47 of system operation; A47: A microbial sample from the anaerobic tank on day 47 of system operation.

**Figure 7 microorganisms-13-01274-f007:**
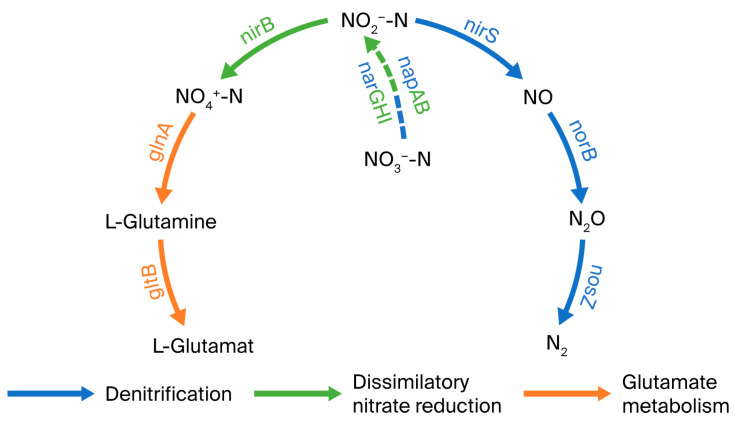
Predicted nitrogen metabolic pathway diagram of strain GH-1.

**Table 1 microorganisms-13-01274-t001:** Inoculation scheme of halophilic denitrifying activated sludge for the aerobic tank.

Stage	Inoculation Time (d)	Inoculation Dosage (m^3^)
Pre-inoculation	/	0
Batch 1	17	5
Batch 2	27, 29, 31	6
Batch 3	38	3
Batch 4	43	5
Batch 5	55	2

The volume of the aerobic tank is 680 m^3^.

**Table 2 microorganisms-13-01274-t002:** Categories of halophilic genes possessed by strain GH-1.

Category	Gene(s)	Halophilic Mechanism
Membrane Regulation Mechanism	cfa (cyclopropane fatty acid synthase)	Catalysis of unsaturated fatty acid cyclopropanation to improve membrane stress resistance [33].
Sodium–Potassium Pump	nhaB (Na^+^/H^+^ antiporter), kdpA/B/C/D/E (high-affinity ATP-driven K^+^ transport system), trkA/H (potassium transporter)	Transportation of Na^+^, K^+^, and H^+^ to maintain intracellular pH and ion balance [30,36,37].
Compatible Solutes	betA/B (betaine–aldehyde dehydrogenase), ectA/B/C/D (ectoine synthase), proA/B/C (proline synthesis)	Compatible solutes (e.g., sugars, amino acids, and their derivatives) regulate cellular osmotic balance [32].
Halophilic Enzymes	phoD (alkaline phosphatase), maeB (malate dehydrogenase)	Alkaline phosphatase and malate dehydrogenase can maintain activity under high-salinity conditions [38,39].

## Data Availability

This article includes the original contributions presented in this study. For further inquiries, please contact the corresponding author. The whole-genome data provided in this study are publicly available in the SRA database of the NCBI (https://www.ncbi.nlm.nih.gov/sra/SRR32491416), with the accession number PRJNA1228538, accessed on 18 March 2025.

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
