# Peer review of "The Aerobic Denitrification Characteristics of a Halophilic Marinobacter sp. Strain and Its Application in a Full-Scale Fly Ash-Washing Wastewater Treatment Plant"

_microorganisms, 2025, doi:10.3390/microorganisms13061274_

Round 1
Reviewer 1 Report
Comments and Suggestions for Authors
The aerobic denitrification characteristics of halophilic Marinobacters p. and -------
- This manuscript showed a fly ash washing wastewater treatment by a halophilic bacteria.
The paper claimed that the nitrogen removal efficiency was higher than those published in previous reports.
- It is skeptical whether the efficiency in this report is significantly higher compared with previous reports because cultural conditions and species of microbes are different.
- From gene analysis, Fig.2 was speculated. However, detection of N2 was not proved. At least, N2 should be suggested from nitrogen balance.
- Three halophilic categories were suggested only by referring previous reports. This is not scientific results, just speculation and discussion matter.
- In full scale operation, gradual increase in N treatment and increase in NH4-N were observed, presumably due to decrease in glutamate metabolism.
- Fig 3 shows specific growth rates in aerobic and anaerobic operations are similar.
However, total N removal levelled off in aerobic operation.
- In full scale operation, initially 10 mg/l NH4 exists and this may be used for cell synthesis. If so, denitrification efficiency should be increased.
Reviewer 2 Report
Comments and Suggestions for Authors
The article presents a characterization study and application of a halophilic bacterial strain in an actual treatment plant treating saline fly ash washing leachate. The authors have done a nice job in both experimentation and presentation of the results. The manuscript is well written and can be published after a few minor revisions.
(1) In Figure 5, the authors can include the control for “unadded bacterial agent stage” throughout the duration of the study instead of 17 days as it is clear that there are large variations of nutrients in the studied influent.
(2) What was the strategy to screen and select this particular microbial species? The authors should explain the isolation section with more clarity. When colonies appeared, what was the strategy to screen colonies and proceed with one specific colony?
(3) The authors should shed more light on the usage of mixed culture for bioaugmentation which could have a better effect on the denitrification process.
(4) In line 135, mention the concentration of bacterial inoculum used for the treatment system.
Reviewer 3 Report
Comments and Suggestions for Authors
-Strain identification: the reference for the primers is lacking.
-What were the criteria for designation at the gender level? Explain and support with references in the manuscript. The supplementary material webpage could not be found.
-Genomic Sequencing section must be supported by references. The authors mention three different sequencing platforms with different types of results, sequencing depths and targets, it is not clear how and for what purpose they were used. There is a methodological information gap in this section that is extremely important to clarify, and this gap is further highlighted in the Results and Discussion sections.
-The microscopy protocol or methods for bacterial observation must be described. What kind of microscope was used?
-Denitrification Characteristics of the Strain: References are needed to support the lab procedures and protocols. How were the nitrogen concentrations determined?
It is not clear the assessment of salt tolerance (0%, 3%, 5%, 7%, 9%, and 12%) since DM medium has 50 g of NaCl. In what kind of physiological information and experimental design are based these experiments? How is the experimental matrix? What kind of a robust experimental design have the authors developed in order to assess all the variables? How many experimental levels were assessed? What were the main experimental variables? Provide reference support.
The authors should be explained how the concentrations of NH4+-N, NO2−-N, NO3−-N, and TN were monitored. References must be considered.
What an A/O treatment system is? Acronims should be describe since the very beginning of the manuscript. The supplementary material webpage could not be found.
Lines 135 to 137: Five liters of pure strain cultures... What was the exact amount of biomass inoculated? (in cell per milliliter, or CFU/ml or OD?
All section. Lines 118 to 147. Given the importance of this information to the manuscript, the methodologies and protocols in the application in full-scale fly ash leachate treatment system section must be described in more technical and scientific detail, e.g. After heavy 121 metal removal, the water is mixed with small amounts of other wastewaters (primarily 122 spray wastewater, acidic gas absorption wastewater, ground flushing wastewater, cooling 123 water, and domestic sewage within the plant area). This paragraph is toogeneral and vague, it is not accurate and the reader loses track of how and to what extent the process is being evaluated. The same for the 2.6. Data Processing Methods.
The table 1 is not clear. How does this table relate to the previous section?
References must be provided for all of Section 3.2
Metabolic mechanisms and processes that are not supported by scientific references are taken for granted.
The authors refer to three different methods on three different sequencing platforms with different findings, depths of sequencing, and targets; the purpose and utility are unclear. There is a critical methodological piece of information missing in this section and it is repeatedly shown as crucial in Results.and Discussion sections. The connection between the title of the paper, the methodological procedures, the results, and the discussion is not clear.
The manuscript must be organized appropriately, depending on the objectives of the work and the methodological processes used to arrive at the results presented and discussed.
Reviewer 4 Report
Comments and Suggestions for Authors
The manuscript entitled: "The Aerobic Denitrification Characteristics of a Halophilic Marinobacter sp. Strain and Its Application in a Full-Scale Fly Ash Washing Wastewater Treatment Plant" presents the results of an extensive work conducted to isolate a bacterial strain capable of denitrification under both, aerobic and anaerobic conditions. The experimental work was designed to produce a robust amount of scientific evidence on the capabilities of Marinobacter sp.
There is no flaw in the methodologies utilized in the present experimental work nor in the results obtained, since technically the work was well done. The results obtained are presented clearly and the discussion is supported by the pool of scientific publications found on this topic. The conclusions are supported by the results.
Although the methodology is in general well structured there is a suggestion I would make. In section 2.5 "Application in Full-Scale Fly Ash Leachate Treatment System", please include the volume of the aerobic tank of the mainstream system in Table 1 (maybe as a footnote). Also, I consider that this volume could be inserted in parenthesis next to the line 142 "aerobic tank of the mainstream system" because in Table 1 the inoculation dosage (m3) presents values up to 6 m3 and it may be a reminder for the reader to see the volume of the aerobic tank. Also, the aerobic tank of the mainstream system works at HRT 6 - 8 h and in order to conduct the study you operated in batch, so at some point may be convenient to mention that you operated indeed in the same facility for which you included the process flow diagram in the supplementary material but you paused temporarly the continuous regime to conduct your research (am I correct?, was the same facility?).
Other than that, I consider that the manuscript can be accepted for publication in this journal.
Round 2
Reviewer 1 Report
Comments and Suggestions for Authors
I ask the authors the complete and refined manuscript. The submitted manuscript is hard to read.
Comments on the Quality of English LanguageAfter the improved manuscript is presented, comments will be given.
Author Response
请参阅附件。

Reviewer 3 Report
Comments and Suggestions for Authors
Many thanks to the authors for taking the time to consider the comments and observations.
The manuscript has been substantially improved.
The information is much clearer and more understandable, the structure is coherent, and the purpose of the research is evident.
Author Response
请参阅附件
